# Sonographic Features of Onychopapilloma: A Single Center Retrospective Observational Study

**DOI:** 10.3390/jcm12051795

**Published:** 2023-02-23

**Authors:** Maria A. Mattioli, Italo F. Aromolo, Cristina B. Spigariolo, Angelo V. Marzano, Gianluca Nazzaro

**Affiliations:** 1Dermatology Unit, Fondazione IRCCS Ca’ Granda Ospedale Maggiore Policlinico, 20122 Milan, Italy; 2Department of Pathophysiology and Transplantation, Università degli Studi di Milano, 20133 Milan, Italy

**Keywords:** onychopapilloma, sonography, ultrasound

## Abstract

(1) Background: Onychopapilloma is a benign tumor of the nail bed and distal matrix. which usually manifests as monodactylous longitudinal eryhtronychia associated with subungual hyperkeratosis. The impossibility to rule out a malignant neoplasm is an indication for surgical excision and histological examination. Our aim is to report and describe the ultrasonographic features of onychopapilloma. (2) Methods: we conducted a retrospective analysis of patients with a histological diagnosis of onychopapilloma who underwent ultrasonographic examination in our Dermatology Unit from January 2019 to December 2021. (3) Results: Six patients were enrolled. Erythronychia, melanonychia, and splinter hemorrhages were the main dermoscopical findings. Ultrasonography detected nail bed dishomogeneity in three patients (50%) and a distal hyperechoic mass (5 patients, 83.3%). Color Doppler imaging did not show vascular flow in any of the cases. (4) Conclusions: the detection of a subungual distal non-vascularized hyperechoic mass by US, together with classical onychopapilloma clinical features, supports the diagnosis, especially in those patients who were unable to perform excisional biopsy.

## 1. Introduction

Onychopapilloma is a benign tumor of the nail bed and the distal matrix that was first described by Baran and Perrin in 1995 [1]. It typically presents as a pink longitudinal band in the nail plate, called longitudinal erythronichia. Distal onycholysis with or without a V-shaped chipping at the free edge of the nail plate, subungual hyperkeratosis, and splinter hemorrhages are other frequent dermoscopical features. Rarely, it can present as melanonychia, leukonychia, or chromonychia. 

Since its recent description, only 316 cases of onychopapilloma have been reported in 25 manuscripts so far. The clinical, dermoscopic, and histopathological features of previously described onychopapillomas are shown in Table 1. This tumor generally occurs in patients in their fifth or sixth decade of life, with some exceptions. However, data on gender prevalence are conflicting. After reviewing the available data extracted from reports in the literature, we identified 113 cases of onychopapilloma in male patients and 183 cases in female patients. In 311 out of 316 cases, the patients received surgery. The histopathological features included papillomatosis, acanthosis, hyperkeratosis, and matrix metaplasia of the nail bed [1,2,3,4,5,6,7,8,9,10,11,12,13,14,15,16,17,18,19,20,21,22,23,24,25]. No ultrasound (US) reports have been published to this date.

## 2. Materials and Methods

We retrospectively reviewed ultrasonographic images of onychopapillomas performed at the Dermatology Unit of Fondazione IRCCS Ca’ Granda from 1 January 2019 to 31 December 2021. The US platform implemented was a Hitachi Arietta V850 with a multifrequency linear array transductor (15.0–18.0 MHz). Transverse and sagittal ultrasonographic images were obtained. 

The study was conducted in accordance with the ethical standards of the responsible committee on human experimentation (institutional and national), with the Helsinki Declaration of 1975, as revised in 2000, and with the Taipei Declaration. All patients provided written informed consent for study participation and publication of photographic material. Because of the retrospective nature of the study, only a notification to the Ethics Committee was requested.

## 3. Results

The epidemiological, clinical, and US findings are summarized in Table 2. Six patients were included in the study, of whom two were males (33.3%) with a medium age of 55.5 (age ranging 23–75). Erythronychia, melanonychia, and splinter hemorrhages were the main clinical findings at onychoscopy (Figure 1). Ultrasonography detected two main features: nail bed dishomogeneity found in three patients (50%) and a distal hyperechoic mass corresponding to the hyperkeratotic spur (five patients, 83.3%) (Figure 2). Color Doppler imaging confirmed the absence of vascular flow in all cases.

## 4. Discussion

Onychopapilloma is a benign nail tumor. The most frequent clinical features are longitudinal erythronychia and subungual hyperkeratosis [1,2,3,4,5,6,7,8,9,10,11,12,13,14,15,16,17,18,19,20,21,22,23,24,25]. Although onychopapilloma is reported as the most common cause of localized longitudinal erythronychia [26], benign tumors, such as the glomus tumor, and malignant neoplasms, such as amelanotic melanoma, Bowen disease, and squamous cell carcinoma, must be considered in the differential diagnosis of monodactylous longitudinal erythronychia in association with a subungual mass. Surgical excision is usually required in all of these cases [27,28]. Clinical and dermoscopic studies could not distinguish between malignancies and histologically confirmed onychopapilloma [11].

On the other hand, melanonychia was reported as a rare presentation of onychopapilloma. Differential diagnosis of longitudinal melanonychia includes the activation of nail matrix melanocytes, benign entities (lentigo or nevus), and malignant entities (melanoma) [29]. Interestingly, Starace et al. recently described six cases of pigmented onychopapilloma and reported that onychoscopy could not rule out melanocytic lesions; therefore, histological examination was necessary [21]. 

However, surgical excision is not without risks to the patient. Delvaux et al. reported their experience with 68 patients with onychopapilloma. Of these, 42% had mild to moderate sequelae (distal fissuring, cicatricial longitudinal erythronychia, punctiform hemorrhages, or onycholysis); while recurrence was observed in 20% of the cases. Four patients were kept on clinical follow-up and did not show any signs of progression [15]. 

US is a valuable tool to aid clinicians in the differential diagnosis of subungual masses [29]. We first reported the US features of six cases of onychopapilloma. In five cases, the presence of a distal subungual hyperechoic mass was identified. No vascular signal was detected by Doppler analysis in any of the examined cases. Moreover, nail bed dishomogeneity was observed in three patients. In our opinion, the nail bed dishomogeneity may represent the US equivalent of the nail bed epithelium acanthosis and matrix metaplasia (Figure 3). The described hyperechoic aspect may histologically correspond to hyperkeratosis. Indeed, compact keratinocytes proliferation US appearance has been described as hyperechoic; for example, in psoriatic scaly plaques [30]. De Berker et al. described how the enhanced transparency of the thinned nail plate and the compression exerted at the margins by the normal nail can act together to generate the reddish band of longitudinal erythronychia [31]. 

Imaging may be helpful in characterizing a subungual lesion [32]: the US appearance of glomus tumors, squamous cell carcinoma, and melanoma usually corresponds to a hypoechoic mass, in contrast to that observed in onychopapilloma [33,34,35,36]. Glomus tumors are described as well-defined and hypervascularized lesions, usually surrounded by a capsule. Squamous cell carcinomas may present an anechoic avascular central zone, corresponding to an area of central necrosis [33]. Moreover, squamous cell carcinomas are characterized by a low-flow vascular signal at Color Doppler that is mostly found at the periphery of the lesion and extends toward the center. Nail unit melanoma is described as an ill-defined hypoechoic mass with intralesional vascularization, which may be difficult to detect in thin lesions. Bony erosions of the distal phalanx may be associated with all three tumors [34,35,36]. 

Surgical excision is recommended for subungual masses that present as painful lesions, typically glomus tumors, or if history and clinical or dermoscopic features are not sufficient to rule out t malignancy [11]. Onychopapillomas may also require surgical treatment due to associated issues such as difficulty picking up small objects (48.2%), pain (40.7%), and distal nail fragility (38.9%) [15]. Clinical follow-up may be indicated in most cases of asymptomatic onychopapilloma. US may be a useful tool to support the diagnosis of onychopapilloma in cases with classical clinical presentation such as longitudinal erythronychia and subungual hyperkeratotic mass. US detection of a subungual distal non-vascularized hyperechoic mass, together with classical clinical features, should represent an indication for clinical and sonographic follow-up, especially in patients unable to perform an excisional biopsy.

However, this case series is too limited and other literary data are scarce; therefore, further studies are needed to determine whether US features may be predictors of malignant versus benign diagnosis. Moreover, the sensitivity of ultrasound is too low to exclude malignancies in a precocious state. 

## 5. Conclusions

US is a non-invasive diagnostic tool that can aid dermatologists in the differentiation of nail tumors presenting with monodactylous longitudinal erythronychia. We are the first to have reported the US features of onychopapilloma, including nail bed dishomogeneity and a distal hyperechoic mass without vascular flow. US may be a useful tool to support the diagnosis of onychopapilloma for the patients who are unable to perform excisional biopsies.

## Figures and Tables

**Figure 1 jcm-12-01795-f001:**
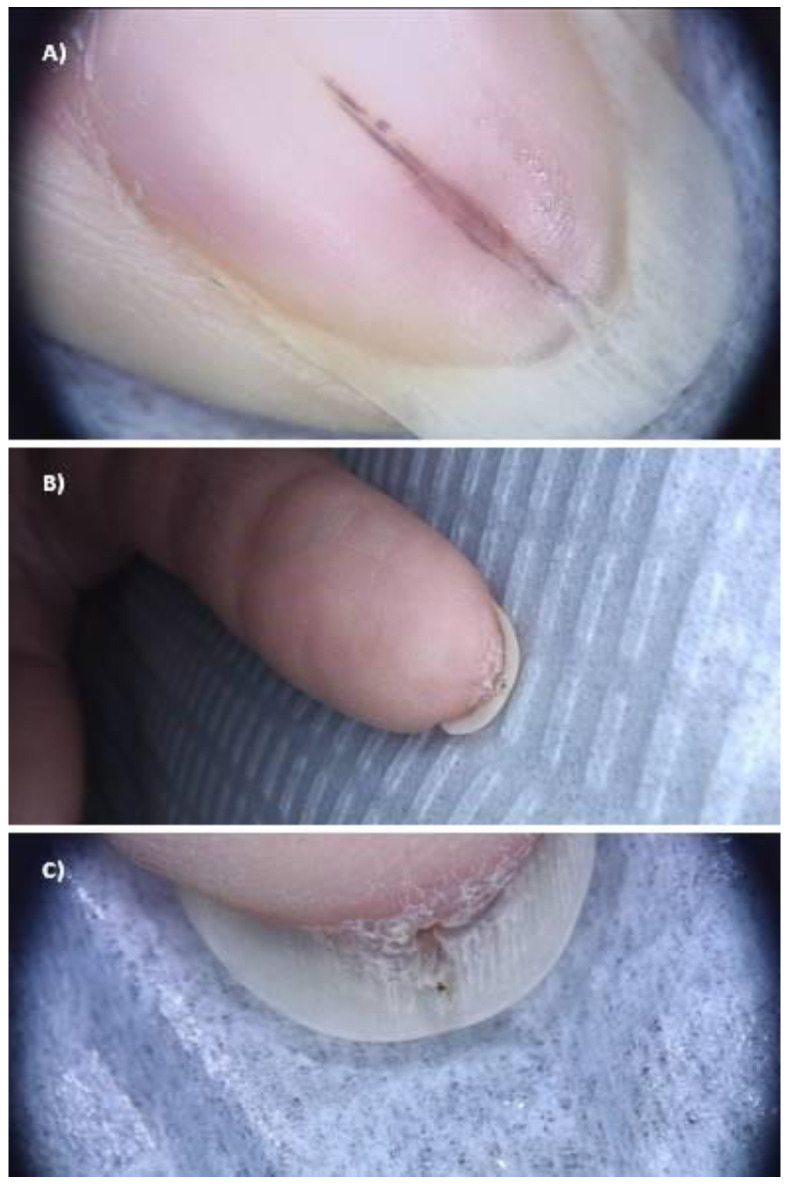
Onychopapilloma presenting with splinter hemorrhages (**A**) and distal subungual keratotic mass (**B**). Dermoscopic appearance of keratotic mass (**C**) (patient no 5).

**Figure 2 jcm-12-01795-f002:**
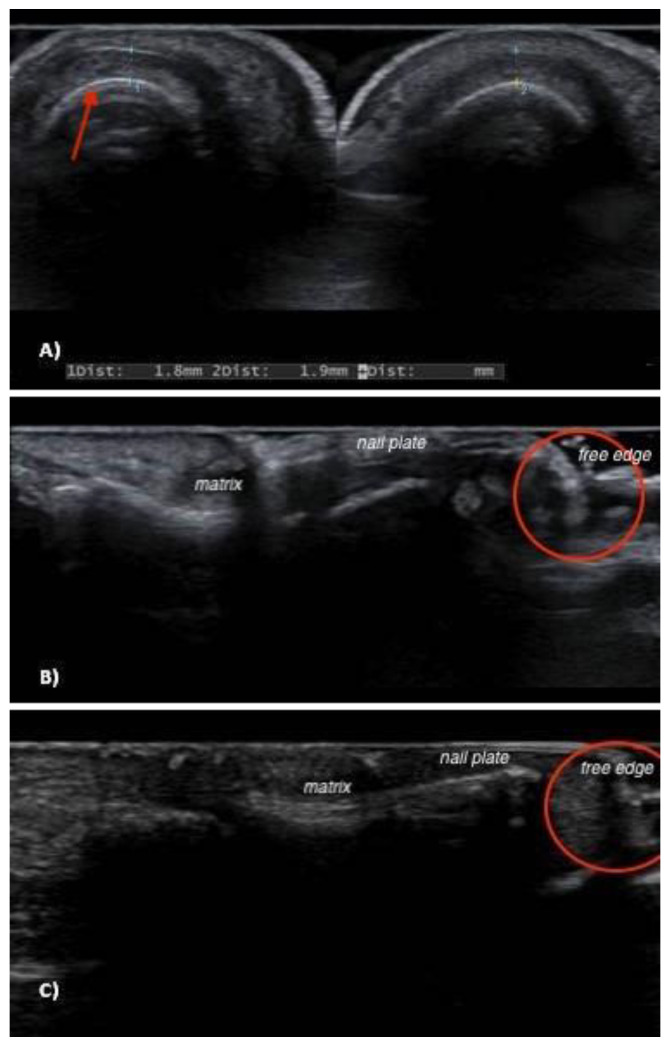
Transversal ultrasonographic sections of the onychopapilloma nail bed (on the left) and a healthy nail bed (on the right) in patient no. 4: there is no difference in thickness, while a slight increase in dishomogeneity (red arrow) can be observed in the onychopapilloma nail bed (**A**). Sagittal ultrasonographic sections of onychopapiloma in patient no. 5 and 6: in both cases, a small hyperechoic mass at the distal end of nail plate can be observed (red circles in (**B**,**C**)).

**Figure 3 jcm-12-01795-f003:**
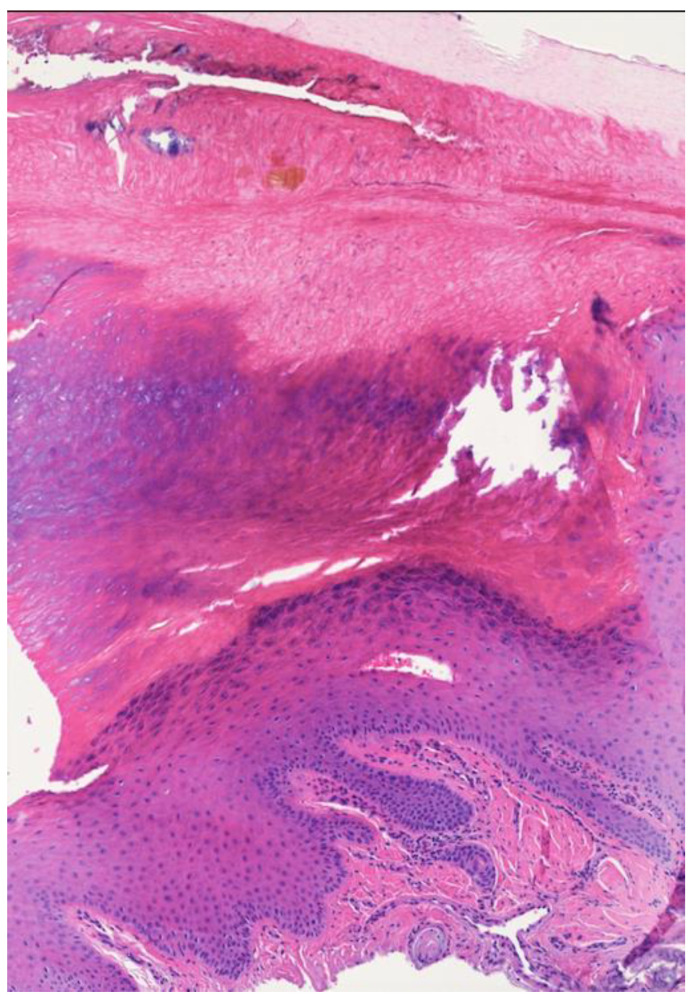
Histopathology of onychopapilloma (patient number 5). The nail bed shows acanthosis and papillomatosis, with some large eosinophilic keratinocytes (matrix metaplasia).

**Table 1 jcm-12-01795-t001:** Previous clinical reports on onychopapilloma in English literature.

Year of Publication	Authors	Patients (n)	Age, Gender (n)	Clinical Features (n)	Histopathological Features	Surgery/Follow-Up (n)	Type of Study
**1995**	Baran and Perrin [1]	4	Not reported	Distal onycholysis (3) Melanonychia (1) Splinter hemorrhages (3) Subungual hyperkeratosis (4)	Nail bed acanthosis, papillomatosis, and matrix metaplasia; multinucleated KC	Surgery	Case reports
**1999**	Baran and Perrin [2]	2	71, F (1), 55, M (1)	Melanonychia (2) Subungual hyperkeratosis (1)	Homogeneous corneal mass with parakeratosis	Surgery	Case reports
**2000**	Baran and Perrin [3]	14	Age Range 18–66 Gender not reported	Distal onycholysis (14) Erythronychia (14) Splinter hemorrhages (14) Subungual hyperkeratosis (14)	Nail bed acanthosis, papillomatosis, and metaplasia; multinucleated KC	Surgery	Case series
**2002**	Gee et al. [4]	1	Not reported	Erythronychia (1) Subungual hyperkeratosis (1)	Nail bed acanthosis, papillomatosis, and matrix metaplasia. Nail matrix basaloid cells with palisading	Surgery	Case report
**2007**	Richert et al. [5]	1	19, F (1)	Distal onycholysis (1) Subungual hyperkeratosis (1)	Nail bed acanthosis and hypergranulosis + lichenoid dermal infiltrate	Surgery	Case report
**2010**	Criscione et al. [6]	1	50, F (1)	Distal V-shaped notch and split (1) Leukonychia (1) Subungual hyperkeratosis (1)	Nail bed acanthosis, papillomatosis, and matrix metaplasia	Surgery	Case report
**2012**	Miteva et al. [7]	1	58, F (1)	Melanonychia (1) Subungual hyperkeratosis (1)	Nail bed acanthosis, papillomatosis, and matrix metaplasia	Surgery	Case report
**2015**	Beggs et al. [8]	1	15, M (1)	Erythronychia (1) Subungual hyperkeratosis (1)	Nail bed acanthosis, papillomatosis, and matrix metaplasia	Surgery	Case report
**2015**	Ito et al. [9]	1	37, F (1)	Distal onycholysis (1) Melanonychia (1)	Nail bed acanthosis and matrix metaplasia. Elongated hyperplastic rete ridges	Surgery	Case report
**2016**	Kim et al. [10]	3	47.8, F (1), M (2)	Erythronychia (1) Multiple yellow chromonychia (1) Reddish-yellow chromonychia (1)	Digitation of the epithelium with abundant eosinophilic cytoplasm	Surgery	Case reports
**2016**	Jellinek et al. [11]	41	58, F (24), M (17)	Erythronychia (41)	Not reported	Surgery	Retrospective single-center study
**2016**	Tosti et al. [12]	47	Age not reported F (33), M (14)	Distal fissures (11) Erythronychia (25) Leukonychia (7) Melanonychia (8) Splinter hemorrhages (11) Subungual mass (47)	Nail bed acanthosis, papillomatosis. Subungual hyperkeratosis and focal parakeratosis	Surgery	Retrospective single-center study
**2017**	Halteh et al. [13]	1	71, F (1)	Leukonychia (1) Subungual hyperkeratosis (1)	Nail bed and matrix metaplasia	Surgery	Case report
**2017**	Sarkissian et al. [14]	1	38, M (1)	Erythronychia (1) Splinter hemorrhages (1) Subungual hyperkeratosis (1)	Nail bed acanthosis, papillomatosis, and parakeratosis	Surgery	Case report
**2018**	Delvaux et al. [15]	68	46.1 ± 2.2, F (42), M (26)	Distal fissures (33) Distal onycholysis (34) Erythronychia (53) Notch in the lunula (39) Subungual hyperkeratosis (56)	Nail bed papillomatosis and acanthosis. Onychogenisis zone	Surgery (63) Follow-up (4) Lost at follow-up (1)	Retrospective single-center study
**2018**	Kim et al. [16]	2	62, 59, F (2)	Distal hyperkeratosis (2) Erythronychia (1) Splinter hemorrhages (2) White- yellow chromonychia (2)	Nail acanthosis, papillomatosis, and matrix metaplasia	Surgery	Case reports
**2018**	Ramos Pinheiro et al. [17]	1	63, F (1)	Erythronychia (1) Hyperkeratotic mass (1)	Nail bed and distal matrix papillomatosis and acanthosis. Hyperkeratotic horn	Surgery	Case report
**2019**	Baltz et al. [18]	1	60, F (1)	Distal onycholysis (1) Erythronychia (1)	Nail bed acanthosis and matrix metaplasia	Surgery	Case report
**2020**	Park et al. [19]	1	50, F (1)	Distal onycholysis (1) Erythronychia (1) Splinter hemorrhages (1) Subungual hyperkeratosis (1)	Nail bed acanthosis and papillomatosis	Surgery	Case report
**2021**	Hashimoto et al. [20]	1	54, M (1)	Melanonychia (1)	Nail bed hyperplasia and matrix metaplasia	Surgery	Case report
**2021**	Starace et al. [21]	6	59, F (3), M (3)	Melanonychia (6)	Distal matrix acanthosis and nail bed papillomatosis	Surgery	Case series
**2022**	Kim et al. [22]	39	Age range 16–77, F (16), M (23)	Erythronychia (22)	Nail bed papillomatosis	Surgery	Retrospective single-center study
**2022**	Liu et al. [23]	11	Age range 13–54, F (6), M (5)	Distal V-shaped notch and split (2) Erythronychia (6) Leukonychia (2) Melanonychia (3) Splinter hemorrhages (2) Subungual hyperkeratosis (11)	Nail bed papillomatosis with or without acanthosis and matrix metaplasia	Surgery	Retrospective single-center study
**2022**	Starace et al. [24]	17	56.3, F (13), M (4)	Distal fissures (8) Distal onycholysis (4) Distal subungual keratotic papule (5) Erythronychia (9) Leukonychia (3) Melanonychia (2) Splinter hemorrhages (2) Yellow-brown chromonychia (1)	Nail bed acanthosis, papillomatosis, and matrix metaplasia. Splinter hemorrhages	Surgery	Retrospective single-center study
**2022**	Yun et al. [25]	50	54.5, F (68%), M (32%)	Distal fissures (12) Erythronychia (11) Subungual hyperkeratosis (15)	Nail bed papillomatosis and matrix metaplasia, subungual hyperkeratosis, and hemorrhage	Surgery	Retrospective single-center study
**Tot (24)**	-	316	F (183); M (113)	Chromonychia (5) Distal fissures (64) Distal onycholysis (59) Erythronychia (183) Leukonychia (12) Melanonychia (22) Notch in the lunula (40) Splinter hemorrhages (34) Subungual hyperkeratosis (151)	Nail bed papillomatosis and matrix metaplasia	Surgery (311), follow up (5)	-

**Table 2 jcm-12-01795-t002:** Epidemiological, clinical, and ultrasound findings.

Patient (Age, Sex)	Clinical Features	Ultrasonography		
		Nail Bed Dishomogeneity	Distal Subungual Mass	Doppler Signal
1 (M, 66)	Melanonychia	NO	YES, hyperechoic	NO
2 (M, 71)	Erythronychia	NO	NO	NO
3 (F, 64)	Melanonychia	YES	YES, hyperechoic	NO
4 (F, 34)	Erythronychia	YES	YES, hyperechoic	NO
5 (F, 23)	Splinter hemorrhages	YES	YES, hyperechoic	NO
6 (F, 75)	Erythronychia	NO	YES, hyperechoic	NO

## Data Availability

The data presented in this study are available on request from the corresponding author. The data are not publicly available due to restrictions for privacy.

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
