# Peer review of "Sonographic Features of Onychopapilloma: A Single Center Retrospective Observational Study"

_jcm, 2023, doi:10.3390/jcm12051795_

Round 1
Author Response
Dear reviewer,
We want to thank you for your kind suggestions to improve our manuscript.
- We have made the corrections as you kindly pointed out to us
- We have checked for any grammar or spelling mistakes.
- An HE figure showing the histology of onychopapilloma has been added
- Table 1 has been summarized as a clinicopathological characterization of onychopapilloma including the patient's age, gender, clinical manifestation, histology, treatment, and follow-up whenever it was possible to extract these data from the corresponding references.
- The differential diagnosis of subungual nevus has been mentioned.
- The conclusion has been modified as " US may be a useful tool to support the diagnosis of onychopapilloma for patients unable to perform excisional biopsies.
- the paper has been modified in order to be more concise and accurate.
Reviewer 2 Report
This is a clearly written article describing 6 cases of onychopapilloma for which ultrasonographic features were observed and described.
In Figure 2A, please indicate on the image where the dishomogeneitiy is found.
In Figure 2B, C please label the sagittal images with anatomic parts of the nail to orient the reader. Where is the nail plate? where is the matrix? It is difficult to tell where the free edge of the nail is and how it differentiates from the subungual lesion.
Please explain in more detail the reason why onychopapillomas are hyperechoic and other subungual neoplasms are hypo- or anechoic. Surely not all SCCs are necrotic leading to central cavitation and a hypoechoic US image.
Author Response
Dear reviewer,
We want to thank you for your kind suggestions to improve our manuscript.
1. In Figure 2A, we have indicated on the image where the dishomogeneity is found.
- In Figure 2B, C we have labeled the sagittal images with anatomic parts of the nail to orient the reader.
- we have explained in more detail the reason why onychopapillomas are hyperechoic and other subungual neoplasms are hypo- or anechoic.
Round 2
Reviewer 1 Report
1. The quality is improved to some extent.
2. Further modification is needed as marked on the text
3. Misspelling is noticed.
4. In discussion part, paragraph 1, 2 and 3 can be combined together to make it more concise. Only differential diagnosis needs to be discussed.
5. Misleading statement as marked in text.
6. A brief summary of the onychopapilloma's clinical features including average age, gender, clinical presentation and treatment can be presented in the introduction part.

Author Response
Dear reviewer,
Thank you very much for your kind suggestions to improve our work.
2. we modified the text according to the notes
3. we reviewed the English spelling
4. In the discussion, we combined together paragraphs 1, 2 and 3
5. We modified the text according to the notes
6. A brief summary of the onychopapilloma's clinical features including average age, gender, clinical presentation and treatment has been added to the introduction part
Best Regards